# Rapid Identification of *Brucella* Genus and Species In Silico and On-Site Using Novel Probes with CRISPR/Cas12a

**DOI:** 10.3390/microorganisms12051018

**Published:** 2024-05-17

**Authors:** Yan Zhang, Yufei Lyu, Dongshu Wang, Meijie Feng, Sicheng Shen, Li Zhu, Chao Pan, Xiaodong Zai, Shuyi Wang, Yan Guo, Shujuan Yu, Xiaowei Gong, Qiwei Chen, Hengliang Wang, Yuanzhi Wang, Xiankai Liu

**Affiliations:** 1College of Food Science and Technology, Shanghai Ocean University, 999 Hucheng Huan Road, Lingang New City, Shanghai 201306, China; 2State Key Laboratory of Pathogens and Biosecurity, Beijing Institute of Biotechnology, 20 Dongdajie Street, Fengtai District, Beijing 100071, China; 3Laboratory of Advanced Biotechnology, 20 Dongdajie Street, Fengtai District, Beijing 100071, China; 4State Key Laboratory for Animal Disease Control and Prevention, Lanzhou Veterinary Research Institute, Chinese Academy of Agricultural Sciences, Chengguan District, Lanzhou 730046, China; 5School of Medicine, Shihezi University, Xinjiang Uygur Autonomous Region, Shihezi 832002, China

**Keywords:** brucellosis, *Brucella abortus*, *Brucella meltisensis*, probes, detection, Cas12a

## Abstract

Human brucellosis caused by *Brucella* is a widespread zoonosis that is prevalent in many countries globally. The high homology between members of the *Brucella* genus and *Ochrobactrum* spp. often complicates the determination of disease etiology in patients. The efficient and reliable identification and distinction of *Brucella* are of primary interest for both medical surveillance and outbreak purposes. A large amount of genomic data for the *Brucella* genus was analyzed to uncover novel probes containing single-nucleotide polymorphisms (SNPs). GAMOSCE v1.0 software was developed based on the above novel eProbes. In conjunction with clinical requirements, an RPA-Cas12a detection method was developed for the on-site determination of *B. abortus* and *B. melitensis* by fluorescence and lateral flow dipsticks (LFDs). We demonstrated the potential of these probes for rapid and accurate detection of the *Brucella* genus and five significant *Brucella* species in silico using GAMOSCE. GAMOSCE was validated on different *Brucella* datasets and correctly identified all *Brucella* strains, demonstrating a strong discrimination ability. The RPA-Cas12a detection method showed good performance in detection in clinical blood samples and veterinary isolates. We provide both in silico and on-site methods that are convenient and reliable for use in local hospitals and public health programs for the detection of brucellosis.

## 1. Introduction

Human brucellosis presents an annual global incidence of approximately 2,100,000 cases among humans [1] and has notable public health repercussions, particularly in numerous developing countries, where prompt and precise diagnoses are of paramount importance. Diagnoses of brucellosis primarily rely on three microbiological techniques: culture, serological tests, and molecular approaches employing a nucleic acid amplification test (NAAT) [2].

Microbiological culture methods serve as the gold standard for clinical diagnosis, primarily through the collection of pathological samples, isolation, and cultivation of bacteria. However, *Brucella* culture methodologies are hampered by an exceptionally prolonged experimental duration, low positive detection rates, and an increased risk for personnel infections and environmental contamination. The clinical diagnosis of brucellosis relies primarily on serologic methods, such as the Rose Bengal test (RBT). However, these methodologies exhibit substantial variability, considering the patient’s medical history, exposure history, antigen recognition, and antibody production pattern. Notably, the “window period” of infection, where antibodies have yet to be produced, limits the diagnostic capability of serology. Furthermore, the presence of cross-antigens between *Brucella* and *Salmonella* undermines the specificity of serological diagnosis for brucellosis [3]. NAATs, particularly polymerase chain reaction (PCR) methods, are currently extensively employed in the diagnosis of *Brucella*. The *omp2*, *omp31*, and *omp28* (*bp26*) genes encoding outer membrane proteins and the insertion sequence IS711 were formerly frequently utilized as target genes in NAATs. Brucellosis is predominantly diagnosed through serological methods in clinical scenarios. The multiplex *Brucella* AMOS PCR assay, which utilizes five primers, has aided in the differentiation of four *Brucella* species, namely *B. abortus*, *B. melitensis*, *B. ovis*, and *B. suis*, and vaccine strains via the IS711 sequence [4]. However, the effectiveness of this sequence is debatable due to its varying sequence and absence in certain strains [2]. The *bcsp31* gene synthesizes an immunogenic membrane protein and thus serves as the most widely employed target for the molecular diagnosis of *Brucella* infection [5,6].

Recently, a group of taxonomists merged the *Brucellae* with the primarily free-living, phylogenetically related *Ochrobactrum* spp. in the *genus Brucella* based on global genomic analysis [7]. Laboratory researchers do not agree with this classification [8], arguing that it is not supported by experts in *Brucella*, and does not take into account the differences in genomic structure, characteristics, and taxonomically related differences between *Ochrobactrum* spp. and *Brucella*. It is necessary to establish an in silico and on-site method to directly distinguish *Brucella* and *Ochrobactrum* spp. The *Brucella* genome presents a high level of conservation, with the genetic similarity across all *Brucella* species exceeding 90% [9]. Moreover, both the physical and genetic maps of the six classical species portray striking similarities, reinforcing their high degree of conservation [10]. Whole-Genome Sequencing (WGS) has the potential for profound discrimination, containing a broader array of target genes and minute base differences that facilitate isolate distinction and outbreak evaluation [11]. The recent influx of available whole-genome sequences from the *Brucella* genus enables the opportunity for in silico reassessment of these loci as potential markers in *Brucella* identification. The swell in genomic sequence data adds complexity to the task of identifying loci with expansive and stable differences. Moreover, whole-genome sequence data have put pressure on many staff members at primary medical institutions. A user-friendly tool for identifying isolate sequences is urgent for them.

As for on-site detection, CRISPR/Cas12a detection methodologies are viewed as a next-generation technology. Their integration with isothermal amplification provides an effective pathway to achieve rapid and accurate detection of bacteria, including *Escherichia coli* and *Staphylococcus aureus*, and SARS-CoV-2 with high sensitivity and specificity [12,13]. The CRISPR/Cas12a detection methodology has several advantages, such as high specificity (SNP detection), convenience (reaction at 37 °C), and programmability (strong scalability of single-stranded DNA). Recombinase Polymerase Amplification (RPA) does not require template denaturation, combined with CRISPR/Cas12a under 37 °C. Integration of the CRISPR/Cas system with RPA has huge application prospects in the field of *Brucella* on-site detection, which does not require precision instruments.

In this study, we aimed to develop reliable methods for the identification of *B. abortus* and *B. melitensis*. As shown in Figure 1, first, a computer-based comparative analysis of the whole-genome data was conducted, and 126 specific probes were uncovered. These probes include *Brucella* genus Probes and *Brucella* species Probes. *Brucella* genus Probes are specific “degenerate fragments” screened from the loci. *Brucella* species Probes are specific fragments screened by small fragments formed after splitting the genome. Then, we developed the publicly available software GAMOSCE v1.0 for the rapid identification of the *Brucella* genus and 5 significant *Brucella* species based on these 126 eProbes. Furthermore, we chose two novel species probes to develop a practical and rapid on-site *B. abortus* and *B. melitensis* detection method that employs the CRISPR/Cas12a detection system combined with RPA. This method has shown good application prospects in clinical blood samples and veterinary isolates.

## 2. Materials and Methods

### 2.1. Data Resources and Initial Genome Screening

The Brucella genus currently comprises 12 species: *B. melitensis*, *B. abortus*, *B. suis*, *B. canis*, *B. neotomae*, *B. ovis*, *B. pinnipedialis*, *B. ceti*, *B. microti*, *B. vulpis*, *B. inopinata*, and *B. papionis* [14]. Genome sequences of these strains were obtained from the GenBank database as of 24 August 2022 (https://www.ncbi.nlm.nih.gov/genome/ accessed on 24 August 2022), including 773 genomes of 11 species of Brucella genus members (Appendix A). Genomic data for *B. papionis* were missing from the NCBI database. To filter out non-compliant genomes, factors such as contamination, strain heterogeneity, and marker lineage were considered using CheckM v1.1.3 software (Figure 1 and Figure 2a) [15]. Consequently, a provisional exclusion of 118 genomes was secured (refer to Appendix A). Later, cluster analysis of 655 whole-genome SNP and core genomes was utilized. The tentative batch of 118 genomes that were initially excused via CheckM was reintegrated based on the preliminary clustering branch. Subsequently, we eliminated some strains with unclear species from our genome set.

### 2.2. Brucella Genus-Specific Fragment Analysis

To find the specific fragments for the *Brucella* genus, we collected 25 loci (listed in Appendix A), including 5s rRNA [16], 16s rRNA [17], *omp22* [18], and *bscp31*. Most were obtained from Multi-Locus Sequence Typing (MLST, 21 loci; https://pubmlst.org/organisms/brucella-spp accessed on 10 January 2024). As shown in Figure 1 and Figure 2b, we employed the local BLAST tool (BLAST-2.7.1+) to probe for fragments of these 25 loci within the genomes of the 773 *Brucella* genus strains. A similar fragment for each locus was compared via Muscle v3.8.31 software. Following this, a Python script was curated to extract the “degenerate fragments” derived from each locus, pursuant to the comparison outcomes. These “degenerate fragments” were then used to interrogate the genetic near neighbors of the *Brucella* genus (as shown in Appendix A), confirming the absence of these “degenerate fragments” within their genomes.

### 2.3. Screening Specific eProbes of Brucella Species

As previous research described [19], the chromosome of *B. melitensis* from strain 16M (GenBank: GCA_000007125.1) served as the reference for the in silico slicing of numerous fragments via a Python script. Leveraging local BLAST (BLAST-2.7.1+), we filtered out fragments present in other *Brucella* species strains that were absent from any *B. melitensis* genome. These fragments were then amalgamated based on fragment overlaps (Figure 1). These fragments, including the SNP site, were unveiled through local BLAST analysis and MEGA-X across all 762 genomes of the *Brucella* genus. The specific fragments corresponding to *B. abortus* and *B. ovis* were procured using a similar strategy. We then retained specific fragments presenting more than one SNP, henceforth referred to as eProbes.

Considering the enormous genomic similarity between *B. suis* and *B. canis*, we constructed a query database using *B. suis* (GenBank: GCA: 000007125.1, renamed Bsuis_001) as the reference sequence. Genome sequences (n = 772) were aligned to this reference sequence, with SNP sites extracted in relation to the genome sequence of each strain. The matrix containing the SNP sites was subsequently processed using Python programming. Subsequently, we screened for SNPs capable of differentiating *B. suis* and *B. canis* from other *Brucella* genus strains, as well as those distinguishing *B. suis* from *B. canis*.

As previous research described [20], the identified SNP, with an upstream and downstream, was utilized to create a 100 bp probe. These probes were scrutinized among *Brucella* genus strains (773 strains) using local BLAST-2.7.1+ software. Based on the elimination of non-specific probes, we procured the probes necessary to differentiate *B. suis* and *B. canis*.

### 2.4. Development and Validation of the Identified Software

The software was created using Python version 3.7 and employed a combination of eProbes sets using nucleotide sequencing data in silico. Whether the data to be tested matched the probe was used to determine whether the sample was positive. If the software identified the sample as being of the *Brucella* genus, it further identified which of the five important *Brucella* species it belongs to. This does not rely on comparison scoring, and the exact answer is provided directly as an output.

We downloaded 1280 *Brucella* genus genomes, including previously tentatively designated *Brucella* that are bacteria that do not belong to the *Brucella* genus, from the BV-BRC website (Bacterial and Viral Bioinformatics Resource Center, https://www.bv-brc.org/, accessed on 11 November 2023), as of 11th November 2023. We excluded duplicate strains from our database, clearly misclassified strains based on whole-genome cluster analysis, and strains only classified into genera, and then 464 genomes remained (Appendix A). These genomes were employed to assess and validate the software.

### 2.5. Detection of B. melitensis and B. abortus Based on Cas12a-RPA

The Cas12a and RPA reactions were performed as previously described [19,20,21]. The reaction process was the RPA amplification product as the substrate of the Cas12a non-specific cutting probe, as shown in Figure 1. The RPA reactions were performed following the instructions of the TwistAmp Liquid Exo Kit (TALQEXO01, TwistDx, Maidenhead, UK), and were conducted at 37 °C for 15 min in a 50 μL volume (Appendix A). RPA reaction components were as follows: Primer A (10 μM) 2.4 μL, Primer B (10 μM) 2.4 μL, 2× Reaction Buffer 25 μL, dNTPs (10 mM each) 2.3 μL, 10× Probe E-mix 5 μL, 20× Core Reaction Mix 2.5 μL, Template 1 μL, H2O 6.9 μL, and 280 mM Magnesium Acetate (MgOAc) 2.5 μL. The target DNA of the Cas12a reaction was provided by the RPA amplification product. The Cas12a reaction was conducted at 37 °C for 30 min in a 20 μL volume (Appendix A). Fluorescence intensities were detected using a Bio-Rad real-time PCR CFX96 instrument in FAM mode (Life Science, Hercules, CA, USA) or with the naked eye under blue light. The sequences of the chosen RPA oligonucleotide primers and crRNAs after pre-experimental analysis using RPA are shown in Table 1.

Lateral flow dipsticks (LFDs; #31203-01; ToloBio, Shanghai, China) were also used to display the detection results. Single-stranded 12-nucleotide DNA probes were modified with FITC and biotin sequences (FITC-5′-GAGACCGACCTG-3′-biotin). Lateral-flow-based immunochromatographic readouts rely on the high affinity of streptavidin and biotin, and FITC binds to gold-nanoparticle-labeled FITC-specific antibodies. The ssDNA probe was excessive compared to the gold particles on the LFDs. When the ssDNA probe is intact, the DNA probe remains bound to the streptavidin line (“C” line) through biotin, creating one color band on the test strip. When the DNA probe is cleaved, FITC and the bound gold-labeled FITC-specific antibodies flow farther on the strip and bind to secondary anti-species antibodies, which leads to the formation of a second color band (“T” line). When the band only appears at line C, the result is negative. When the band appears at the T-line position, the result is positive. The absence of any strip is invalid. The Cas12a reaction product (10 μL) with 40 μL of NEB 3.0 buffer was placed in a PCR tube and incubated with an LFD strip (5 min). The Cas12a reaction was conducted using a 20-pmol ssDNA probe.

### 2.6. Clinical Samples and Vaccine Strains’ Acquisition and Extraction

To avoid biosecurity risks, the extracted genomes of the following vaccine strains were used to test these probes in the field: *Brucella melitensis* vaccine strain M5-90Δ26 was a deletion of the *bp26* gene after 90 passages, and *Brucella melitensis* vaccine strain M5 was developed in China from the virulent strain *B. melitensis* M28 and used to vaccinate sheep and goats [22]. Some scholars speculate that the widely used Chinese *B. abortus* vaccine strain A19 was derived from *B. abortus* vaccine strain S19 (attenuated and isolated by Buck in 1923) before 1956 [23]. *Brucella suis* vaccine strain S2 was developed in China through serial subculturing on media, and *B. abortus* 104 M has been used as a vaccine strain in humans against brucellosis for six decades in China [24]. *Brucella melitensis* 5134 and 5321 were isolated from sheep in Nilek County, Ili Area, Xinjiang Autonomous Region (unpublished).

As described in previous research, genetically distant neighbors *Bacillus anthracis* A16R, *B. cereus* BC307, and *B. subtilis str.* 168 [21], and near neighbors *Ochrobactrum anthropic* CICC21622 and *Ochrobactrum intermedium* CICC20571 (China Center of Industrial Culture Collection, CICC), were used as negative controls.

Clinical blood samples were collected from a high-risk group of people in a beef and mutton processing enterprise in northwest China. Informed consent was obtained prior to sample processing. All samples were stored at −80 °C in a biorepository until processing. Genomic DNA from clinical blood was extracted using automatic nucleic acid extraction instrument with reagents (TGuide S16; #DP348; Tiangen, Beijing, China,), according to the manufacturer’s instructions.

### 2.7. Ethics Approval and Consent to Participate

All relevant ethical guidelines were followed, and any necessary IRB and/or ethics committee approvals were obtained. Informed consent was obtained from the participants prior to sample processing. Protocols were approved by the Ethics Committee of the First Affiliated Hospital, Shihezi University School of Medicine (approval number KJ2022-156-01).

### 2.8. Polymerase Chain Reaction

The genomic DNA of blood samples was tested using the PCR method outlined in the literature to evaluate the specificity and effectiveness of the RPA-Cas12a method. The primers of *bscp31* [5,6] and the experimental steps used in PCR were all performed in accordance with the literature.

## 3. Results

### 3.1. Initial Screening of the Brucella Genus Genome Database

As of 24 August 2022, the NCBI database comprised 773 genomes spanning 11 species within the *Brucella* genus (excluding *B. papionis*). This mainly encompassed 261 *B. abortus* whole-genome sequences, 356 *B. melitensis* sequences, 81 *B. suis* sequences, and 29 *B. canis* sequences (Appendix A). All of these genomes were utilized as genomic databases for specific SNP screening. For ease of description, we systematically renumbered the downloaded genomes, incorporating species names for clearer delineation (Appendix A).

To ensure dependable construction of the genome library, an initial evaluation of the downloaded genomic data quality is imperative (Figure 2a). Tentative batches of 118 genomes were temporarily excluded from our database using CheckM v1.1.3 (Appendix A) [15]. Subsequently, we employed a cluster analysis using 655 whole-genome SNP and core genomes to identify and exclude 9 *Brucella* genomes with imprecise classifications from our preliminary database [25,26]. The remaining 646 strains were then categorized into branches corresponding to their respective species. The process involved the re-categorization of the temporarily excluded 118 genomes based on the preliminary clustering branch, enabling the successful infiltration of all but 2 genome strains into their appropriate species branch. Consequently, we removed 11 strains with ambiguous species from the strain set, thus retaining 762 strains from the *Brucella* genus for additional SNP analysis (Appendix A).

### 3.2. Screening of Loci for Identifying the Brucella Genus

To find fragments that could be used to identify the *Brucella* genus, we evaluated 25 loci (Appendix A) on the chromosome. After analysis, we found eight specific fragments of loci: AGenus-BCSP31, AGenus-cobQ, AGenus-ddlA, AGenus-fumC, AGenus-glk, AGenus-mutL, AGenus-putA, and AGenus-trpE (Figure 1 and Figure 2b; Appendix A). These “degenerate fragments” containing SNPs were named *Brucella* genus eProbes (BGPs).

### 3.3. Screening, Calibration, and Validation of Brucella Species-Specific SNPs

To find fragments suitable for *B. melitensis* identification, the chromosomes of *B. melitensis* strain 16 M were digitally sliced into numerous fragments (Figure 2c). Following a process of elimination, integration, and alignment, we discovered 1084 fragments inclusive of the SNP site, based on the *Brucella* strain library. Furthermore, we constructed 33 *B. melitensis* eProbes (BMPs), each containing SNP sites and possessing GC contents ranging from 40 to 60% (Appendix A). Adopting the same approach, we accrued 28 *B. abortus* eProbes (BAPs) and 28 *B. ovis* eProbes (BOPs) (Appendix A).

Nevertheless, considering the extreme genomic similarities between *B. suis* and *B. canis*, it is very difficult to use a one-step approach to distinguish *B. suis* from all other *Brucella* species. This prompted the idea of employing a two-step method for *B. suis* identification (Figure 2d). As described above, we conducted high-throughput comparisons. Ultimately, in step one, we were able to discern 14 *B. suis* and *B. canis* eProbes (BSCPs) in setA (BSCPs_setA) that discriminated *B. suis* and *B. canis* from other members of the *Brucella* genus. Then, in step two, 15 BSCP eProbes in setB (BSCPs_setB) were chosen to differentiate between *B. suis* and *B. canis* (Appendix A).

We retrospectively analyzed the genomes of 773 *Brucella* genus strains to determine the specificity and effectiveness of 5 eProbe sets (Figure 1). Notably, none of the probes were incorrectly identified. We next analyzed the genomes of 464 newly added *Brucella* strains (Appendix A) from the website and 10,271 strains, including *Bacillus, Acinetobacter*, and *Yersinia* genera, as test sets (Appendix A) to calibrate and validate the applicability of the 5 eProbe sets. These probe sets accurately identified the *Brucella* genus, *B abortus*, *B. melitensis*, and *B. suis* (Figure 2e). These results suggest that these eProbe sets can be used to distinguish the *Brucella* genus and four *Brucella* species that can harm humans from their genetic near-neighbors *Ochrobactrum anthropi* and *Brucella intermedia*. Unfortunately, the BOPs sets were not further assessed since there were no more added *B. ovis* genomes available.

### 3.4. Development of Identification Software Based on These Probes

Encouraged by the specificity and effectiveness of these probe sets, we compiled software for the rapid identification of the *Brucella* genus and five significant *Brucella* species with genome sequence data. The software was named *Brucella* Genus, Abortus, Melitensis, Ovis, Suis, Canis genome-based identification with E-probe (GAMOSCE v1.0). As long as one of these probes was exactly matched, the result was identified as positive. This software can be downloaded by anyone for free from: https://github.com/844844/GAMOSCE accessed on 5 March 2024, for Windows users (and can also be downloaded from Appendix A of this manuscript). The identification of strains can be completed with one click by inputting complete sequence data (draft genome sequence compatible) on a Windows-based PC.

After successful completion of the software compilation, the 11 deleted *Brucella* strains from our database (Appendix A) were identified using GAMOSCE v1.0 scanning (Appendix A). The results showed that Babortus_204, Babortus_231, Babortus_250, and Bsuis_065 were *B. melitensis* (Figure 3a). Babortus_223, Babortus_230, Binopinata_003, Bmelitensis_138, and Bmelitensis_309 were *B. suis*. Bmelitensis_307 was not *B. melitensis* but rather *B. abortus*. Because our software could only identify five significant species, *Brucella* genus Bsuis_080 was not determined here.

To verify these results, we combined these 11 strains with 5 *B. abortus* strains, 5 *B. melitensis* strains, 5 *B. suis*, and 3 *B. inopinata* strains (one newly added from NCBI’s latest update) as a validation set for whole-genome SNP phylogenetic analysis. The whole-genome SNP cluster was consistent with that of GAMOSCE. Babortus_204, Babortus_231, Babortus_250, and Bsuis_065 clustered with *B. melitensis*, and Babortus_223, Babortus_230, Binopinata_003, Bmelitensis_138, and Bmelitensis_309 clustered with *B. suis* after whole-genome phylogenetic analysis (Figure 3b). Bmelitensis_307 clustered with *B. abortus*, and Bsuis_080 clustered with *B. inopinata*. The 11 strains are marked in red in Figure 3b. These results indicate that the 11 strains in the NCBI genome database had been misidentified, even though their genome sequences were complete.

### 3.5. Application of Brucella Identification Software

We collected 203 genomes (Appendix A) from *B. abortus*, *B. melitensis*, *B. suis*, and *B. canis* isolates in previously published literature [27,28,29,30,31]. These genomes were used as a test set to further evaluate the software. As shown in Figure 3c, the results identified with GAMOSCE were consistent with those in the literature.

Furthermore, we examined the genomic information of a strain isolated (GenBank: GCA_018604785.1) from an abortion storm on a dairy farm in India. Previous research [32] suggested that this isolate was identified as *B. abortus* via the amplification of the *Brucella abortus omp28* gene. However, the isolate was identified as *B. melitensis* using GAMOSCE v1.0 software (Figure 3d and Appendix A). The strain was confirmed to cluster more homologously with *B. melitensis* than with *B. abortus* in the whole-genome phylogenetic analysis (Figure 3e). These findings indicate the high precision and efficiency of GAMOSCE as a tool for identifying the *Brucella* genus and five significant *Brucella* species.

### 3.6. Detection of B. abortus and B. melitensis Based on SNP Sites by RPA Combined with Cas12a

Having confirmed that these eProbe sets efficiently differentiated the *Brucella* genus and five significant *Brucella* species in silico, we developed an RPA combined with CRISPR/Cas12a assay to facilitate the rapid and specific detection of mainly human pathogenic species of *B. melitensis* and *B. abortus* with a naked-eye readout and lateral flow assay (Figure 1). As previously described in Materials and Methods section, the principle of lateral flow-based immunochromatography with RPA-Cas12a was shown in Figure 4a. Because *B. melitensis* and *B. abortus* are the most prevalent agents of brucellosis in humans, accounting for over 95% of total cases [33], and *B. melitensis* is associated with the most severe disease manifestations, genomic DNA was used as the detection substrate. Their DNA concentrations were adjusted to 10^6^ copies/μL (Appendix A). For the specific detection of *B. melitensis* or *B. abortus* genomic DNA, CRISPR RNAs (crRNAs) corresponding to the probes of *B. melitensis* or *B. abortus* were designed and evaluated (Appendix A).

The two probes (BAP03 and BAP13) were selected with the ability to distinguish *B. abortus* A19 via a naked-eye readout under blue light and a lateral flow assay from closely related species (*B. melitensis* M5-90Δ26 and *B. suis* S2), as well as other negative control species, including *Ochrobactrum anthropi*, *Ochrobactrum intermedium*, *B. anthracis* A16R, *B. cerues* 307, and *B. subtilis* 168 (Figure 4b). To ascertain the detection threshold, the genomic DNA of A19 was diluted across a range from 10^6^ to 10^0^ copies/μL (Figure 4c; Appendix A). The crRNA targeting the SNPs (BAP03) exhibited higher sensitivity and was capable of detecting concentrations as low as 10^3^ copies/μL of genomic DNA (Figure 4c).

Similarly, probes BMP14 and BMP31 enabled the distinction of *B. melitensis* M5-90Δ26 from neighboring bacteria (A19 and S2), as well as from other control species (Figure 4b). The genomic DNA of M5-90Δ26 was also diluted from 10^6^ to 10^0^ copies/μL (Appendix A). The crRNAs for the target SNPs (BMP31) demonstrated high sensitivity and detected a minimal concentration of 10 copies/μL of genomic DNA (Figure 4c). Consequently, BAP03 and BMP31-crRNA were selected for the screening assays of *B. abortus* and *B. melitensis*.

### 3.7. Performance Evaluation an d Application of the RPA-Cas12a Assay Using Clinical Samples

Following the establishment of the RPA-Cas12a assay, which employed novel probes for discriminating between *B. melitensis* and *B. abortus*, we applied this method to detect nucleic acid samples extracted from new veterinary isolates, as well as blood samples collected from clinical cases.

Two new veterinary isolates, 5134 and 5321, were identified as *B. melitensis* using GAMOSCE (Figure 5a and Appendix A). They were also identified as *B. melitensis* using BMP31-crRNAs (Figure 5b), exhibiting the same fluorescence signals and lateral flow dipstick bands as M5-90Δ26. The BAP03-crRNAs demonstrated the precise and direct detection of *B. abortus* 104 M and 2308 through both fluorescence and lateral flow methods (Figure 5c). These results aligned with the identification made through AMOS (Abortus Melitensis Ovis Suis)-PCR (Appendix A). The AMOS-PCR products were further sequenced to confirm the species.

Subsequently, we assessed a dataset comprising 37 clinical blood samples from a high-risk population prone to brucellosis, which identified 21 positives for brucellosis by the RBT in serum (Appendix A). These blood samples were extracted (Appendix A), amplified in RPA reactions, and processed further as Cas12a reactions. Using these novel crRNA probes, we detected 21 positive *B. melitensis* cases and 0 positive *B. abortus* cases from these samples (Figure 5d,e). However, these clinical blood samples could not be distinguished using AMOS-PCR (Appendix A). The PCR with target *bscp31* products, as confirmed by sequencing, was subsequently unblinded to assess the precision of the RPA-Cas12a assay. The *bscp31* PCR results showed that 11 of the clinical samples were positive (Figure 5f and Appendix A for the original figure). The result distribution between these two assays is depicted in Figure 5g. The distribution of samples with positive results was consistent for both methods. RPA-Cas12a was shown to improve detection sensitivity in the initial experiments. The distribution of RBT results in the serum was completely consistent with RPA-Cas12a with the BMP31-crRNA assay. The results of RPA-Cas12a with the BMP31-crRNA assay were confirmed by RPA product sequencing (Appendix A).

## 4. Discussion

In this study, we analyzed the genomes of 773 *Brucella* genus strains acquired from the NCBI database, developed GAMOSCE v1.0 software, and evaluated the specificity and effectiveness of specific probes in silico via different genomics databases. GAMOSCE correctly identified all *Brucella* genus strains, demonstrating a strong discrimination ability. Our results showed that the development of GAMOSCE was important to accurately identify different *Brucella* species after sequencing the sample. As shown in Figure 3, eleven *Brucella* strains from the NCBI database were incorrectly classified, although they all completed whole-genome sequencing. The isolate from an abortion storm on a dairy farm in India was also incorrectly identified as *B. abortus* (actually *B. melitensis*) [32]. The identification results of the GAMOSCE v1.0 software were consistent with those of the whole-genome phylogenetic analysis. These instances again showed the power and convenience of our software.

Whole-genome analyses can generate greater discrimination by containing more target genes. Core genome MLST has many advantages, including high-resolution typing of outbreaks, a rapid and simple analysis workflow, and facilitating the sharing of sequencing results between (inter)national laboratories [34,35]. cgMLST schemes for several other bacterial species have been established and evaluated as more genome sequence data have become available [35,36]. Notably, cgMLST was recently employed for high-resolution typing of outbreaks with *Brucella* strains [34]. These methods are all good ways to distinguish strains, but they require skilled bioinformatics professionals. Using genome sequence data, GAMOSCE can be used to identify the *Brucella* genus from near-neighbor bacteria in silico based on probe sets in seconds. With the wider application of whole-genome sequencing in the field of public health, our GAMOSCE tool will become increasingly useful.

The current study also has some limitations. The genomes of *B. ovis* in the three databases (NCBI (https://www.ncbi.nlm.nih.gov/datasets/genome/?taxon=234, accessed on 24 August 2022), BV-BCR (https://www.bv-brc.org/view/Taxonomy/234#view_tab=genomes, accessed on 20 November 2023), and JGI (https://gold.jgi.doe.gov/organisms?Organism.Organism+Name=brucella&setColumns=yes, accessed on 23 November 2023)) were duplicated and mainly derived from the article published by Alvarez et al. [37]. The genomics data we used to construct BOPs were from the NCBI database. It was not possible to find new genomes of *B. ovis* to assess the specificity of the BOPs.

Compared to other pathogens [38,39,40,41], the development of CRISPR-based diagnostic methods for *Brucella* is currently lagging. CRISPR-based diagnostic assays have been previously demonstrated for the detection of *Brucella* by Dang et al. [42] and Xu et al. [43]. Their studies used classical detection targets and did not focus on *Brucella* species’ differentiation. Additionally, variable temperatures are not suitable for use in the field. However, the accurate identification of *B. abortus* and *B. melitensis* is crucial for epidemiological investigation and disease control. In this study, we developed an RPA-Cas12a assay for the direct detection of *B. abortus* and *B. melitensis* at a constant temperature (37 °C) for 45 min using two highly specific targets screened for GAMOSCE v1.0 software.

Finally, we evaluated these new protocols using two veterinary isolates and clinical blood samples. As shown in Figure 5g, our protocols in blood samples achieved an improved detection sensitivity in the initial experiments and could be used to distinguish *Brucella* species. As mentioned above, the IS711 element is prone to variation and deletion in some strains. AMOS-PCR could not accurately determine the species in the blood samples. Therefore, stable SNP detection methods for species identification are very important. Of course, the nucleic acid amplification assay is strongly dependent on high-quality nucleic acids extracted from clinical samples.

The first line of defense for controlling infectious diseases is to apply fast, sensitive, and accurate diagnostic methods, which will provide clinical doctors with information to better make preprocessing clinical and management decisions. Our GAMOSCE v1.0 software can accurately and rapidly distinguish the *Brucella* genus and five significant *Brucella* species. It will be helpful for microbiologists, medical doctors, and clinical laboratories to identify *Brucella*. The probes contained in the software pave the way for the development of microbiological diagnostic kits or chips. Brucellosis is a disease that is particularly relevant in low- and middle-income countries. The sequencing capacity of most public health laboratories is limited. GAMOSCE v1.0 software was designed to be compatible with draft genome sequences. The RPA-Cas12a assay with LFDs would thus enable rapid identification of *B. abortus* and *B. melitensis* to support outbreak investigation and public health containment efforts.

## 5. Conclusions

*Brucella* is the etiologic agent of brucellosis. Due to the high homology between *Ochrobactrum anthropic*, *Ochrobactrum intermedium*, and *Brucella* genus, and the isolation of *Ochrobactrum* from patients, taxonomists have attempted to combine them into a single genus. This idea is controversial among clinical and environmental microbiologists, and it is urgent to establish a method for their differentiation. The 773 genomes of Brucella genus strains were retrospectively analyzed to determine the 126 specificity and effectiveness eProbes, which divided into 5 eProbe sets. GAMOSCE software based on these 126 novel eProbes was validated on different Brucella datasets and correctly identified all Brucella strains, demonstrating a strong discrimination ability. The RPA-Cas12a detection method based on newly screened probes detected a minimal concentration of 10 copies/μL and showed good performance in detection in clinical blood samples and veterinary isolates. The RPA-Cas12a assay with LFDs and an incubator (under 37 °C) is a simple and reliable method for use in local hospitals for the detection of *B. abortus* and *B. melitensis*.

## Figures and Tables

**Figure 1 microorganisms-12-01018-f001:**
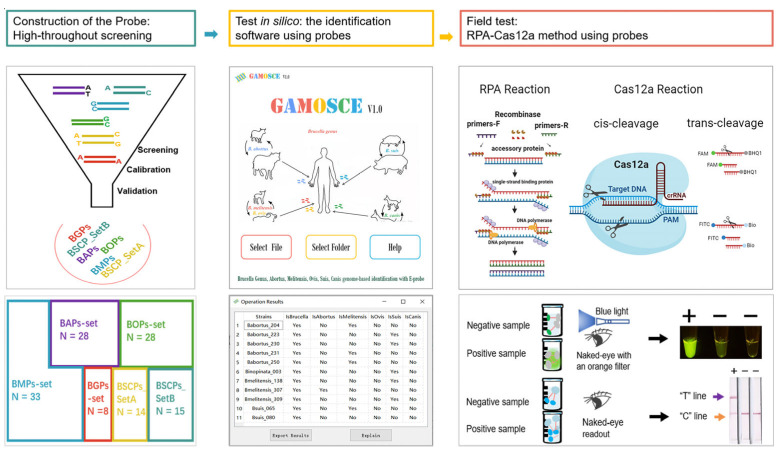
Development of in silico and field testing identification methods for the *Brucella* genus and 5 significant *Brucella* species. BGPs, *Brucella* genus eProbes; BAPs, *Brucella abortus* eProbes; BMPs, *Brucella melitensis* eProbes; BOPs, *Brucella obvis* eProbes; BSCPs, *Brucella suis* and *canis* eProbes; RPA, Recombinase Polymerase Amplification.

**Figure 2 microorganisms-12-01018-f002:**
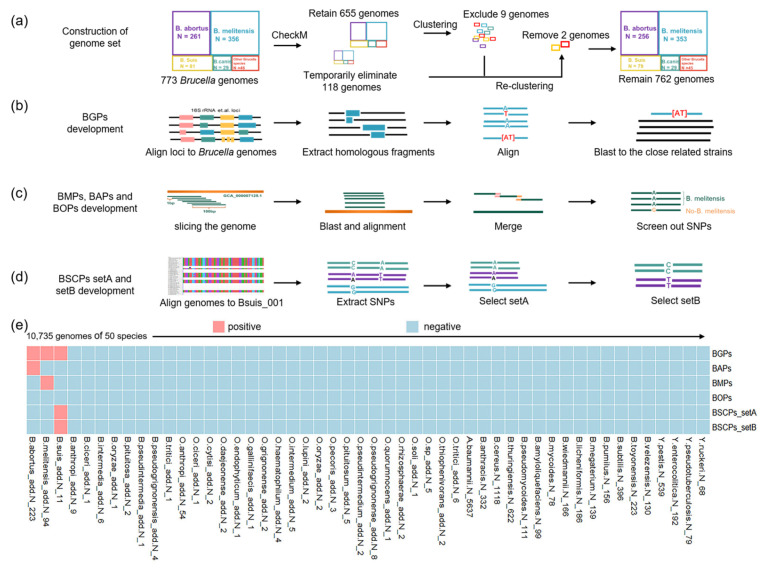
Schematic diagram and evaluation of probe set development. (**a**) Construction of the genome library. (**b**) *Brucella* genus eProbes (BGPs) are specific “degenerate fragments” selected from the loci, which can distinguish the *Brucella* genus containing 11 species from the closely related strains. (**c**) *B. abortus* eProbes (BAPs), *B. melitensis* eProbes (BMPs), and *B. ovis* eProbes (BOPs) are specific fragments screened by small fragments formed after splitting the genome and can differentiate the respective strains from genetic near neighbors. (**d**) *B. suis* and *B. canis* eProbes setA (BSCPs-setA) are specific probes used to discriminate *B. suis* and *B. canis* from other strains, which are constructed by the SNPs screened by the reference genome and other strains forming a gene sequence alignment array. BSCPs-setB are specific probes used to discriminate *B. suis* from *B. canis*. (**e**) These probes showed accuracy, as assessed by a library of more than 10,000 strains from both the website and the local database.

**Figure 3 microorganisms-12-01018-f003:**
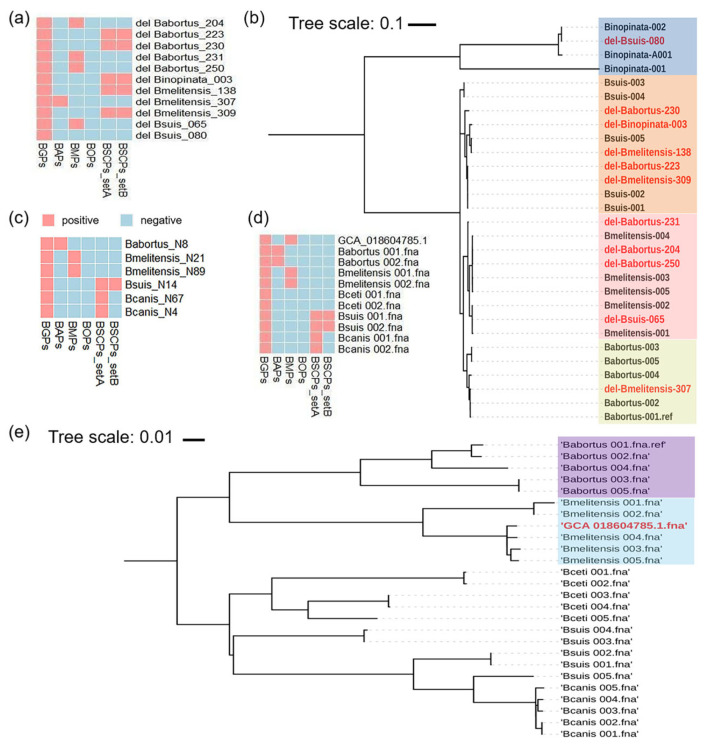
Display of GAMOSCE v1.0 software analysis for identification of the *Brucella* genus and 5 significant *Brucella* species. (**a**) The results of the 11 *Brucella* strains deleted by GAMOSCE v1.0 scanning were different from the original identification in the NCBI database. The corresponding table of the assembly identifiers (IDs) in this study is shown in Appendix A. (**b**) The results of GAMOSCE v1.0 were proven via whole-genome SNP analysis. (**c**) The results of software identification for these different strains are consistent with those in the literature. (**d**) Strain GCA_018604785.1 was identified by software as *B. melitensis* rather than *B. abortus*, which was confirmed by whole-genome SNP (**e**) cluster analysis. The corresponding information on strains from the literature is shown in Appendix A.

**Figure 4 microorganisms-12-01018-f004:**
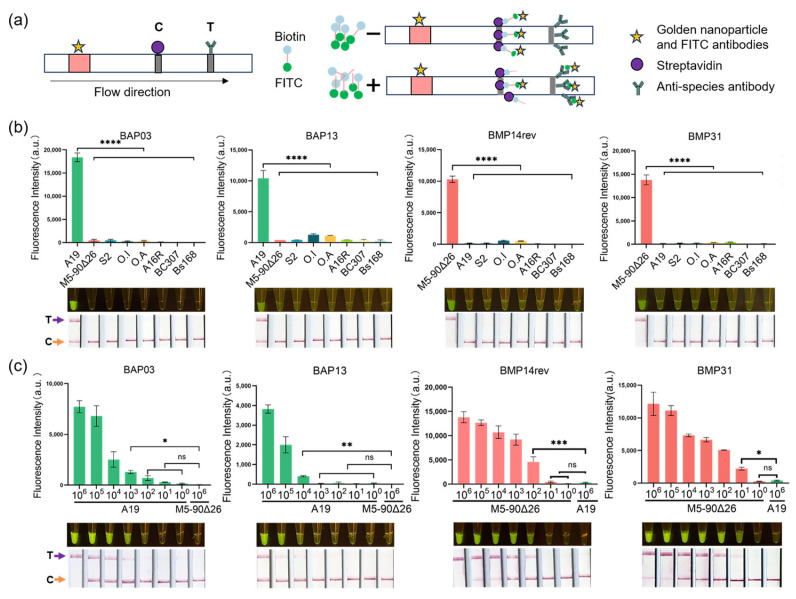
RPA-Cas12a for *B. abortus* and *B. melitensis* detection with fluorescence and lateral flow. (**a**) The principle of lateral flow-based immunochromatography with RPA-Cas12a. When the band only appears at line C, the result is negative. When the band appears at the T-line position, the result is positive. The absence of any strip is invalid. Specificity assay (**b**): the *B. abortus*-specific crRNAs corresponding to the two probes can distinguish *B. abortus* (strain A19) from *B. melitensis* (strain M5-90Δ26), *B. suis* (strain S2), the closely related genera (*Ochrobactrum anthropic* and *Ochrobactrum intermedium*), and *Bacillus* spp. (strain A16R, BC307 and Bs168) within 45 min. The *B. melitensis*-specific crRNAs corresponding to the two probes can distinguish *B. melitensis* from other species (one-way repeated-measure analysis of variance). Sensitivity assay using the two crRNAs to detect A19 and the two crRNAs to detect M5-90Δ26 (one-way analysis of variance) (**c**). ****, *p* ≤ 0.0001; ***, *p* ≤ 0.001; **, *p* ≤ 0.01; *, *p* ≤ 0.05; ns, not significant. Negative sample: The “T” line shows no color, the “C” line is present in lateral flow, and no fluorescence is observed. Positive sample: The “T” line color is present in the lateral flow, and strong fluorescence is observed. O. A., *Ochrobactrum anthropi*; O. I., *Ochrobactrum intermedium*; a.u., Arbitrary unit of fluorescence intensity, by Bio-Rad real-time PCR CFX96.

**Figure 5 microorganisms-12-01018-f005:**
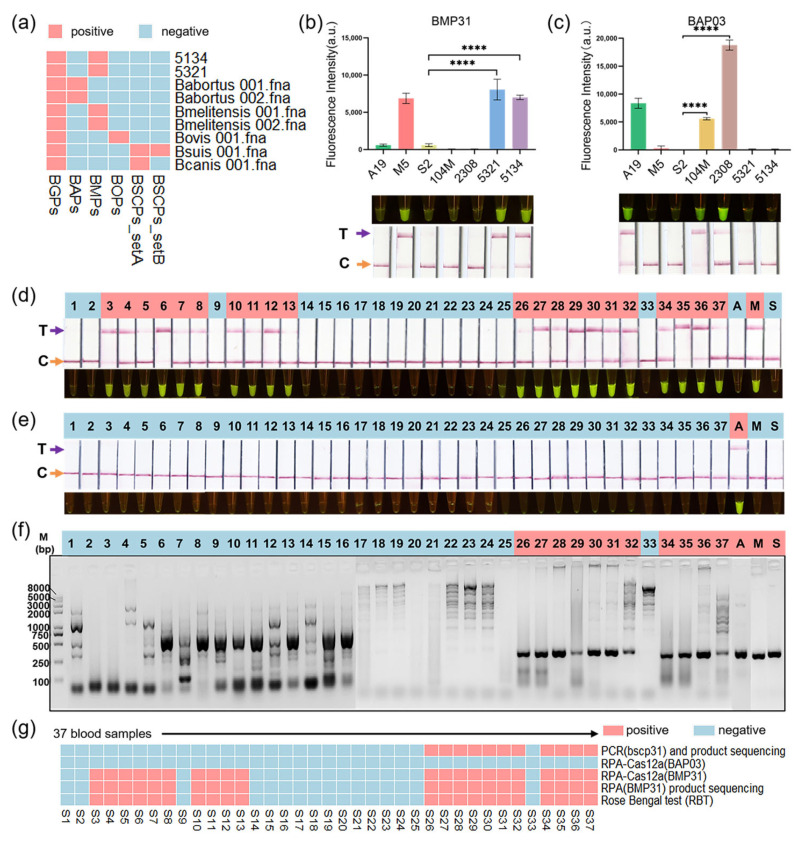
Testing of the RPA-Cas12a assay on 2 new isolates and a validation set of 37 clinical samples. Clinical samples were processed according to a standard clinical laboratory workflow with no biological or technical replicates. The two new isolates, 5134 and 5321, were identified as *Brucella melitensis* using GAMOSCE (**a**) and the corresponding crRNA of BMP31 (**b**). When the band only appears at line C, the result is negative. When the band appears at the T-line position, the result is positive. The absence of any strip is invalid. ****, *p* ≤ 0.0001. (**c**) The corresponding crRNA of BAP03 could distinguish between *B. abortus* 104 M and 2308. These clinical samples were identified via RPA-Cas12a with BMP31-crRNA (**d**) and BAP03-crRNA (**e**) and PCR with target bscp31 (**f**). (**g**) Comparison of PCR with target bscp31, RPA-Cas12a assay with the two probes, and Rose Bengal test (RBT) in clinical samples. M and M5, *B. meltensis* M5-90Δ26; A, *B. abortus* A19; S, *B. suis* S2.

**Table 1 microorganisms-12-01018-t001:** Information on RPA primers and crRNAs used in this study.

Probes	crRNA Sequence (5′-3′)	Primers	Primer Sequence (5′-3′)
BMP14	AAUUUCUACUGUUGUAGAUGUGAAUGUGCCUUCGCA	BMP14rev-F4	TGCCCGGTTTTCAAGCTTTTGC**TTTG**GTG
		BMP14rev-R	TCAAGGATGCGGATGTGAACTGGCGCA
BMP31	AAUUUCUACUGUUGUAGAUAAAUAACACGGGCCACC	BMP31-F5	ACAATTGGCCGCAGCCCGCGCACTC**TTTC**AAAT
		BMP31-R	AGGCAGGCTATCGCGCTGTTCAGAAAGCATATT
BAP03	AAUUUCUACUGUUGUAGAUAUUUCCGAUCAGGCCAG	BAP03-F4	AGAACGGTTACGGCCGCTTGAGGA**TTTTT**ATT
		BAP03-R	ACGGATAGGTGCTTCTTCCAGATTTTCCGCCT
BAP13	AAUUUCUACUGUUGUAGAUCAGGCCGCCUGUCGUUC	BAP13-F4	CTCGTCAAAGCTTTGGTTTCATC**TTTA**CAG
		BAP13-R	GAAATTCAAGGTTTACCAGCATATCGGCGAT

Bold text indicates the PAM sequence that we used amplification primers to introduce into the amplified product. Underlined parts indicate the positions of the complementary sequences.

## Data Availability

Data are contained within the article or Appendix A.

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
