# Peer review of "Rapid Identification of Brucella Genus and Species In Silico and On-Site Using Novel Probes with CRISPR/Cas12a"

_microorganisms, 2024, doi:10.3390/microorganisms12051018_

Round 1
Reviewer 1 Report
Comments and Suggestions for Authors
Line 450-1: there is no clinical importance of differentiating between abortus and melitensis, they are treated identically and have roughly similar clinical presentation
Line 477-8: the ochrobactrum controversy is different than described here. The problem is when you have an innocent ochrobactrum and categorize it as brucella, which is almost never innocent. But did the authors prove that ochrobactrums are accurately categorized as innocent by this method?
Line 71-3: again, correct reference on the ochrobactrum issue but no description of the actual controversy.
Line 59-60: cross-reactivity with salmonella needs referencing
In the conclusions, a comment on the feasibility and affordability of the wide use of the method is needed.
Author Response
Dear Reviewer
Thank you for your kind consideration of our manuscript entitled “Rapid identification of Brucella genus and species in silico and on-site using novel probes with CRISPR/Cas12a” (Paper #microorganisms-2945346). We also thank the reviewers for their positive comments and constructive suggestions, which we have used to improve the quality of our manuscript.
Please find the comments with our point-to-point responses attached. The corresponding amendments have been highlighted in the revised manuscript. We hope both you and the reviewers will be satisfied to find that the revised manuscript meets the standards required for publication in Microorganisms.
Please see the attachment.Thank you again for your work during this period. We hope you are safe and healthy and that your work is going smoothly. We look forward to hearing from you.
Yours sincerely,
Xiankai Liu

Reviewer 2 Report
Comments and Suggestions for Authors
In this review the authors evaluate Brucella genomes, identify disparate fragments, confirm genome similarities by bioinformatic analysis, and develop novel probes using CRISPR/Cas12a for specific detection of Brucella abortus and B. melitensis in a lateral flow assay. The data suggests a great deal of sophisticated work on the Brucella genome. I have the following comments for the authors to consider:
1) I would suggest changing the title to indicate that the novel probes used CRISPR/Cas12a techniques.
2) On line 72, the reference should be Hordt et al 2020. The #6 reference is a publication disputing the Hordt recommendation of combining Brucella and Ochrabacterum. Also, I believe there is a typo in the sentence in line 72.
3. On line 92 the definition of RPA should be defined. I did not note it previously in the manuscript.
4. Beginning on line 94, I believe the authors should more clearly state that a big portion of the data reported in the manuscript was to genomic analysis of 25 loci in available Brucella genomes, identifying disparate fragments, confirming genomic relationships, and that data was used to develop Cas12a probes for B. abortus and B. melitensis. After reading the introduction I was somewhat confused by the initial work since it expanded into identifying disparate fragments and probes to differentiate multiple Brucella strains.
5. The results section reports detection of fluorescence intensity in au units but the methods on lines 168-182 do not define hos this measurement was taken. For this reviewer, lateral flow devices are generally read visually so the measurement of fluorescence intensity should be defined.
6. One thing I remained confused on was the specifics of the amplification reaction prior to running the PCR products on the lateral flow containing the probes. Was the AMOS-PCR assay conditions and primers used or was something else used. The results section should be modified to make this clear to readers.
6. I thought all of the figures could be simplified to make them more readable. In many instances, parts of figures are so small as to be unreadable. I noticed that parts of the figures are not cited in the text. For example, I found 4a and 4b are cited on line 350, 4c and 4d on line 357, and 4e and 4f are cited on line 350-351. 4 g, 4h, and 4i are not cited. I believe there is an error in the citation of figures in this paragraph as I believe 4f and 4g should be cited on lines 351-352 (not 4e and 4f) and 4h and 4I should be cited on lines 358. Again, aspects of figures 4 and 5 are so small that it is difficult to determine what is being demonstrated.
I had concerns regarding figure 5 as in looking at 4d, I detect 14 positive lateral flow responses to the B. melitensis Cas12a probe, but the authors report 21 on line 402. The authors also report that 11 were positive on the bscp31 PCR reaction. One question I had was how the authors confirmed the accuracy of the additional positives on the B. melitensis Cas12a probe rather than their argument that they were negative on the AMOS-PCR assay. Did they try other PCR assays to confirm that a Brucella target could be detected. If the AMOS-PCR was use for amplification, why was there no PCR product detected on gels, but detection did occur on the lateral flow device? Otherwise, what is the confirmation that the additional positives are not false positives. Also, addressing how my interpretation of 4d suggests fewer than 21 positives should be addressed. Some of the lanes in 4d appear to be negative for the control band which the legend would suggest makes that test inaccurate? I cannot eliminate the possible that resolution of figures has eliminated visible bands. Many subgraphs within this figure are hard to distinguish. I believe in figures 4a, 4d and 4e the C on the Y axis means control and suspect the T stands for target. However, this is not defined.
Author Response
Dear Reviewer
Thank you for your kind consideration of our manuscript entitled “Rapid identification of Brucella genus and species in silico and on-site using novel probes with CRISPR/Cas12a” (Paper #microorganisms-2945346). We also thank the reviewers for their positive comments and constructive suggestions, which we have used to improve the quality of our manuscript.
Please find the comments with our point-to-point responses attached. The corresponding amendments have been highlighted in the revised manuscript. We hope both you and the reviewers will be satisfied to find that the revised manuscript meets the standards required for publication in Microorganisms.
Please see the attachment. Thank you again for your work during this period. We hope you are safe and healthy and that your work is going smoothly. We look forward to hearing from you.
Yours sincerely,
Xiankai Liu
